# Elucidation of Prebiotics, Probiotics, Postbiotics, and Target from Gut Microbiota to Alleviate Obesity via Network Pharmacology Study

**DOI:** 10.3390/cells11182903

**Published:** 2022-09-16

**Authors:** Ki-Kwang Oh, Haripriya Gupta, Byeong-Hyun Min, Raja Ganesan, Satya Priya Sharma, Sung-Min Won, Jin-Ju Jeong, Su-Been Lee, Min-Gi Cha, Goo-Hyun Kwon, Min-Kyo Jeong, Ji-Ye Hyun, Jung-A Eom, Hee-Jin Park, Sang-Jun Yoon, Mi-Ran Choi, Dong Joon Kim, Ki-Tae Suk

**Affiliations:** Institute for Liver and Digestive Diseases, College of Medicine, Hallym University, Chuncheon 24252, Korea

**Keywords:** gut microbiota, obesity, equol, isoflavone, *Lactobacillus paracasei JS1*, IL6

## Abstract

The metabolites produced by the gut microbiota have been reported as crucial agents against obesity; however, their key targets have not been revealed completely in complex microbiome systems. Hence, the aim of this study was to decipher promising prebiotics, probiotics, postbiotics, and more importantly, key target(s) via a network pharmacology approach. First, we retrieved the metabolites related to gut microbes from the gutMGene database. Then, we performed a meta-analysis to identify metabolite-related targets via the similarity ensemble approach (SEA) and SwissTargetPrediction (STP), and obesity-related targets were identified by DisGeNET and OMIM databases. After selecting the overlapping targets, we adopted topological analysis to identify core targets against obesity. Furthermore, we employed the integrated networks to microbiota–substrate–metabolite–target (MSMT) via R Package. Finally, we performed a molecular docking test (MDT) to verify the binding affinity between metabolite(s) and target(s) with the Autodock 1.5.6 tool. Based on holistic viewpoints, we performed a filtering step to discover the core targets through topological analysis. Then, we implemented protein–protein interaction (PPI) networks with 342 overlapping target, another subnetwork was constructed with the top 30% degree centrality (DC), and the final core networks were obtained after screening the top 30% betweenness centrality (BC). The final core targets were IL6, AKT1, and ALB. We showed that the three core targets interacted with three other components via the MSMT network in alleviating obesity, i.e., four microbiota, two substrates, and six metabolites. The MDT confirmed that equol (postbiotics) converted from isoflavone (prebiotics) via *Lactobacillus paracasei JS1* (probiotics) can bind the most stably on IL6 (target) compared with the other four metabolites (3-indolepropionic acid, trimethylamine oxide, butyrate, and acetate). In this study, we demonstrated that the promising substate (prebiotics), microbe (probiotics), metabolite (postbiotics), and target are suitable for obsesity treatment, providing a microbiome basis for further research.

## 1. Introduction

Obesity is an serious health issue globally because it is related to diverse diseases, such as diabetes, atherosclerosis, hypertension, heart attack, and even cancers [1]. The main cause of obesity a persistent between consumed energy and expended energy for a long period of time [2]. A criterion used to assess the severity of obesity is body mass index (BMI), with BMI values of 30.00 or more indicating obesity [3]. At present, obesity is prevalent in all ages, with the worldwide prevalence of obsesity expected to increase to 573 million by 2030 [4]. 

Common medications administered for short-term weight management include phentermine and diethylpropion as appetite suppressants [5]. Orlistat is approved for long-term oral administration, acting as a pancreatic lipase inhibitor to interrupt dietary fat absorption [6]. However, these drugs are associated with negative side effects, such as headache, nausea, dry mouth, constipation, diarrhea, and even anxiety [7]. 

Recently, a report demonstrated that metabolites from the gut microbiota can exert favorable efficacy to ameliorate metabolic disorders, including obesity [8]. Another report suggested that metabolites produced by the gut microbiota act as regulators to maintain energy balance in host system [9]. The gut microbiota is related to the etiology of obesity and its associated metabolic disorders, for instance, by regulating the fermentation of dietary polysaccharides, fat consumption, and even obesity [10]. In addition, a study showed that utilization of prebiotics, probiotics, and synbiotics (the mixture of prebiotics and probiotics) may affect the production of chemical messengers (hormones and neurotransmitters) and inflammatory elements, interrupting diet intake stimulators that result in obesity [11]. Therefore, favorable dietary food (prebiotics) and gut microbes (probiotics) may exert positive effects on obesity. As mentioned above, prebiotics (a precursor of postbiotics) converted into postbiotics (defined as metabolites) via probiotics (known as gut microbiota) have been documented as critical constituents for the treatment of obesity; however, their core targets have not been completely elucidated in highly complicated microbiome systems. Thus, we pioneered potential prebiotics, probiotics, postbiotics, and more importantly, key targets to establish the four key components against obesity.

Flavonoid-abundant foods, such as fruits and vegetables, exert therapeutic effects, including anti-inflammation, antioxidant, hypertension, and anti-obesity actions, a relationship that is expounded by characteristics of the gut microbiome to a certain extent [12]. Flavonoid are considered significant nutritional constituents in therapeutics due to their favorable physicochemical properties, providing an improved absorption rate, increased therapeutic capacity, and fewer adverse effects relative to other compounds [13]. The gut–liver axis is a cross junction between the gut, its microbial colonization, and the liver, regulating the signaling pathways tuned by nutritional, genetic, and environmental variables [14]. Therefore, flavonoids with advantageous pharmacokinetic characteristics for use as agents against some diseases, including obesity, might be promising additives produced by the gut microbiota.

Additionally, the construction of multiple biological networks offers prospective insight to elucidate complex pharmacological information, such as microbial interactions, protein–protein interaction (PPI) network analysis, and even topological analysis [15,16]. Specifically, biological network models can serve as a paradigm to uncover the underlying causes of complex diseases [17]. Network pharmacology is an integrated analytical methodology to pioneer significant components (bioactives, proteins, diseases, and genes) [18]. With the development of bioinformatics, network pharmacology can decode the mechanism of action in complex biological systems through interdisciplinary studies, suggesting that network pharmacology is a convergent approach to shift from “one target, one compound” to “multiple targets, multiple compounds” [18,19,20].

We constructed a microbiota–substrate–metabolite–target (MSMT) network to reveal the underlying therapeutic values of the interactions. A previous report suggested that *Lactobacillus paracasei JS1*, isoflavone, and equol might be significant components in the treatment of skin and intestinal disorders [21]. Another significant factor related to obesity is IL6. This target is a key element exerting pharmacological efficacy against obesity and is better known as a targeted to inhibit obesity [22]. 

Our description shows that important components, such as prebiotics, probiotics, and postbiotics, can dampen obesity in the microbiome and bed targeted via network pharmacology analysis. Targets with high betweenness centrality (BC) (the route value of the shortest path between nodes) can be used to identify significant nodes (or targets) in the network [23]. Key targets with highest BC values are noteworthy therapeutically relevant candidates that can be used to treat diverse diseases [24]. Hence, we generated PPI networks based on the BC values of each target, and targets with high connectivity values in the networks were considered therapeutic targets against obesity.

Despite insufficient data to elucidate a crucial therapeutic underpinning of obesity etiology, our findings contribute to unraveling the potentiality of an antiobesity effect in complex microbiome systems.

## 2. Hypothesis

We postulated that targets with a high degree of betweenness centrality (BC) can serve as therapeutic candidates against obesity. Targets with a high correlation degree value in PPI networks based on a high BC value are potential targets for the treatment of obesity. The most stably bound metabolites on a key target in the molecular docking test (MDT) were considered key postbiotics, and microbes produced by postbiotics were defined as key probiotics.

## 3. Methods and Materials

Biologically substantial datasets with abundant data on the association between ligands and targets enable researchers to employ network pharmacology as an efficient method for drug discovery or development (Table 1). Such web-based datasets are available freely to users to obtain valuable information that can be applied to microbiome and network pharmacology. In this study, we implemented new approach to explore the complex microbiome and its network via public databases and network pharmacology strategies.

The metabolites generated by the gut microbiota were identified via gutMGene (http://bio-annotation.cn/gutmgene/) (accessed on 31 May 2022) [25]. We adopted the similarity ensemble approach (SEA) (https://sea.bkslab.org/) (accessed on 31 May 2022) [26] for mining analysis and SwissTargetPrediction (STP) (http://www.swisstargetprediction.ch/) (accessed on 31 May 2022) [27] to search for targets linked to the metabolites. Obesity-responsive targets were identified by DisGeNET (https://www.disgenet.org/) (accessed on 1 June 2022) [28] and OMIM (https://www.omim.org/) (accessed on 1 June 2022) [29]. Crucial targets were utilized to identify the interaction between each node through PPI networks. Then, we constructed MSMT networks for descriptive purposes. Finally, MDT was implemented to evaluate the binding stability between metabolites and targets. The study protocol was conducted as follows.

Step 1: Retrieval of metabolites produced by gut microbiota through gutMGene. The metabolites converted by gut microbes were identified by a subfolder in the downloads section (http://bio-annotation.cn/gutmgene/public/res/gutMGene-human.xlsx) (accessed on 31 May 2022) in gutMGene v1.0, suggesting key metabolites reported to date.

Step 2: Targets associated with the metabolites were mined by SEA and STP databases. The metabolites were selected in simplified molecular input line entry system (SMILES) format to load in the two databases, the format of which was converted by PubChem (https://pubchem.ncbi.nlm.nih.gov/) (accessed on 31 May 2022). Obesity-related targets were identified via DisGeNET (https://www.disgenet.org/) and OMIM (https://www.omim.org/).

Step 3: Selection of intersecting targets between SEA and STP. We selected the intersecting targets to achieve rigor and exactness from the two databases. In detail, an SEA database was constructed by Dr. Shoichet’s group to identify the affinity of compound targets and eventually reveal their binding stability [26]. Additionally, 23 of 30 drug targets suggested by SEA were experimentally confirmed [30]. STP was used as a tool to predict targets for cudraflavone C, a species of flavanols, which were experimentally demonstrated [31]. Thus, we utilized the two databases to enhance the success rate in this study. With the help of VENNY 2.1 (https://bioinfogp.cnb.csic.es/tools/venny/) (accessed on 1 June 2022) Venn diagram, we selected the overlapping targets. 

Step 4: Identification of crucial targets among obesity-related targets and the overlapping targets extracted by SEA and STP. We considered intersecting targets to be crucial targets. 

Step 5: The crucial targets from Step 4 were analyzed by the String database version 11.5 (https://string-db.org/) (accessed on 1 June 2022) [32], and we adopted PPI networks to identify relationships using the R package.

Step 6: Construction of sub-PPI networks with the highest degree centrality (DC) values in the upper 30% from the PPI networks (Step 5) via R package. DC is determined as the number of edges on each node [33].

Step 7: Construction of a subnetwork with the highest betweenness centrality (BC) values in the top 30% from the sub-PPI networks (Step 6) via R package. BC is a measurement of the influence of node in a network [34]. The highest BC value reflects the relative significance of aspects of biological effect in the networks, i.e., nodes with higher BC values have increased potential therapeutic value against disease [23].

Step 8: Description of MSMT networks via R package. The most important elements against obesity were indicated by the size degree of the circle. 

The size of each component (node) describes the number of interactions (edge) in the MSMT networks.

The microbiota, substrate, metabolite, and target were merged to describe their relationships in Microsoft Excel in combination with R package to identify their connectivity. We defined specific compounds of prebiotics as substrates (S) that are pre-metabolites before becoming primary metabolites in the MSMT networks. 

Step 9: Initial screening was performed by MDT, with a cutoff less than -6.0 kcal/mol or the lowest Gibbs energy in each complex. The metabolites were downloaded in .sdf format from PubChem (https://pubchem.ncbi.nlm.nih.gov/) and converted to .pdb format via Pymol. The selected .pdb format was converted to .pdbqt format to implement the MDT. The crystal structure of each target was obtained by RCSB PDB (https://www.rcsb.org/) (accessed on 2 June 2022). The MDT was implemented to verify the affinity between metabolites and targets via AutoDockTools-1.5.6 [35]. 

The docking box size was solidified with x = 40 Å, y = 40 Å, and z = 40 Å. The active site of the crystal structure was formatted with a cubic box in the center: AKT1 (x = 6.313, y = -7.926, z = 17.198) and IL6 (x = 11.213, y = 33.474, z = 11.162). Both hydrophilic and hydrophobic interaction analyses were performed with LigPlot + 2.2. (https://www.ebi.ac.uk/thornton-srv/software/LigPlus/download2.html) (accessed on 3 June 2022) [36]. 

Step 10: Drug-resemblance and toxicity properties were validated by SwissADME (http://www.swissadme.ch/) (accessed on 4 June 2022) and the ADMETlab web-based tool (https://admetmesh.scbdd.com/) (accessed on 4 June2022) [37]. These two factors are critical elements to facilitate new agents; thus, we assessed their physicochemical values and side effects. The study workflow is represented in Figure 1.

## 4. Results

A total of 208 metabolites were obtained via the gutMGene database and identified by the SEA (1256) and STP (947) databases (Appendix A). A total of 668 targets overlapped between SEA and STP (Figure 2A); a Venn diagram plotter program intersected 342 common targets between the 668 targets and obesity-related targets (3028) (Figure 2B) (Appendix A).

In the PPI networks, NMUR2, PAM, BRS3, UTS2R, and SSTR4 did not exhibit any interactions with other targets and consisted of 337 nodes and 4492 edges (Appendix A). The subnetwork was obtained by selecting the upper top 30% in terms of degree centrality (DC) (Table 2), comprising 106 nodes and 1441 edges (Appendix A). 

After screening the top 30% in terms of betweenness centrality (BC) of the subnetwork, which comprised 32 nodes and 254 edges (Table 3) (Figure 3).

In the BC subnetwork, the targets with the top three BC values were albumin (ALB), interleukin-6 (IL6), and AKT serine/threonine kinase 1 (AKT1), which were considered the core targets connected with microbes to alleviate obesity. The gut microbes directly related to the production of metabolites were identified as *Escherichia coli*, *Lactobacillus paracasei JS1*, *Eubacterium limosum*, and *Enterococcus durans M4-5*; however, there are additional unknown microbes (Unknown 1, Unknown 2, Unknown 3, and Unknown 4) that produce favorable metabolites against obesity, as indicated in the MSMT networks (Figure 4). Additionally, the beneficial prebiotics to convert into metabolites against obesity were identified as tryptophan and isoflavone, which can produce indole and equol via *Escherichia coli* and *Lactobacillus paracasei JS1*, respectively [21,38]. Likewise, there are unknown prebiotics (Unknown 5, Unknown 6, Unknown 7, Unknown 8, and Unknown 9) that may act against obesity. However, the information on prebiotics, probiotics, and postbiotics (Unknown 10) has yet to be confirmed, although ALB was selected as a core antiobesity target. 

Five metabolites (equol, 3-indolepropionic acid, trimethylamine oxide, butyrate, and acetate) on IL6 (PDB ID: 4NI9) and one metabolite (indole) on AKT1 were selected to perform MDT (Table 4). We observed that the equol–IL6 complex (-7.4 kcal/mol) (Figure 5) docked most stably, indicating its promise as a postbiotic and a target. 

Then, the drug-likeness properties and toxicity of equol were evaluated by the SwissADME and ADMETlab platforms according to Lipinski’s rule of five, including criteria of molecular weight (≤500), H-bond acceptor (≤10), H-bond donor (≤5), MlogP (≤4.15), bioavailability score (>0.1), and topological polar surface area (TPSA) (<140). Our results indicate that equol can be accepted by pharmacokinetics parameters to be assessed as a new agent (Table 5).

Despite an acceptable therapeutic value, an agent may not be an end product due to unexpected toxicity. Therefore, a drug candidate should exceed the limits of toxicity for further verification. Accordingly, equol was evaluated in terms of hERG blockers, rat oral acute toxicity, eye corrosion, and respiratory toxicity, including LD50 (5.238 mg/kg), via the ADMETlab platform (Table 6). Our observational study suggests that Isoflavone as a prebiotic, *Lactobacillus paracasei JS1* as a probiotic, equol as a postbiotic, and IL6 as a target might exert positive effects on obesity.

## 5. Discussion and Conclusion

Previous studies have demonstrated that the metabolites from the gut microbiota act as significant agents in a wide range of diseases, such as cancer, stroke, irritable bowel syndrome, and even mental disorders, including obesity [39,40]. In this study, we analyzed microbiota–substrate–metabolite–target (MSMT) networks and found that *Lactobacillus paracasei JS1*, isoflavone, equol, and IL6 exhibited considerable connectivity in the networks, indicating significant antiobesity effects. 

Isoflavone is a major compound isolated from soybeans; its pharmacological action has been established against cancers, osteoporosis, metabolic disorders, and neurodegenerative symptoms [41]. Furthermore, equol, as an isoflavone-derived metabolite, has diverse favorable therapeutic effects on human health, such as estrogenic and antioxidant efficacy [42]. An animal test demonstrated that equol can result in a reduction in body weight, white adipose tissue, and depression caused by dietary restriction [43]. More importantly, its therapeutic effects can be confirmed in the context of equol produced by gut microbes. In contrast, in patients who cannot produce equol due to a lack of equol-producing gut microbes, an alternative is to directly administer equol with pharmaceutical forms [44]. 

A recent report showed that *Lactobacillus paracasei JS1* can convert isoflavones into equol via fermentation [45]. Another report demonstrated that equol treatment in collagen-induced arthritis (CIA) inhibited the expression level of IL6 and its receptor at the point of rheumatoid arthritis (RA) [46], suggesting that IL6 is a key regulator of inflammatory levels. The considerable production of IL6 from adipocytes may lead to metabolic disorders, such as obesity; moreover, IL6 cytokine signaling in adipose tissue is associated with hepatic insulin resistance and steatosis [47]. IL6 spontaneously stimulates the secretion of free fatty acid (FFA) by exerting a negative effect on glucose metabolism [47]. Therefore, IL6 inhibition might be an optimal therapeutic target against obesity. 

We conducted analysis to identify a key target via topological analysis based on betweenness centrality (BC). In drug network analysis, a drug with a high BC value tends to be associated with several therapeutic applications, with considerable promising for treatment of diverse diseases [36,48]. In particular, a study demonstrated that targets with top 30% BC related to Xiao-Chai-Hu-Tang (Chinese herbal formula) were selected to uncover the mechanism of action against colorectal cancer, providing a theoretical basis for clinical tests [49]. Based on this result, we adopted “top 30% BC” as a threshold in this study.

Network pharmacology research is a powerful tool to monitor targets, pathways, drugs, and diseases in light of the rapid development of databases [50]. We constructed a stepwise workflow to investigate key targets and metabolites to treat obesity by combining public databases. Bioinformatics can be used not only to efficiently mine for drug candidates but also to facilitate drug repurposing [51]. Furthermore, Huangqin decoction was proven an antidiabetic enteritis agent via the combination of network pharmacology and gut microbiota sequencing [52]. 

Accordingly, we performed a network pharmacology study to evaluate the pharmacological value of the key target identified by a microbiome study. We conducted an observational trial to explicitly elucidate a key target against obesity in complex microbiome networks by reporting up-to-date information. According to the MSTM network results, we selected five potential metabolites and three targets for MDT, with results indicating that equol can bind stably to IL6, which suggests that equol may ameliorate obesity by inhibiting IL6. 

Taken together, our results show that isoflavone (prebiotic), *Lactobacillus paracasei JS1* (probiotic), equol (postbiotic), and IL6 (Target) are the most crucial components against obesity in current microbiome research. However, accumulation of information concerning the microbiome is subject to some limitations. Due to the limitations of bioinformatics and cheminformatics, we suggest that further preclinical or clinical test should be conducted to specify the four identified elements. 

## Figures and Tables

**Figure 1 cells-11-02903-f001:**
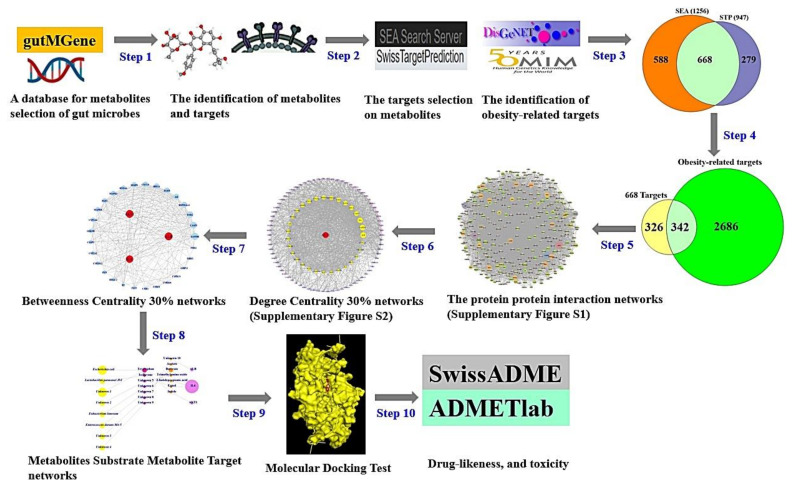
The workflow of this study.

**Figure 2 cells-11-02903-f002:**
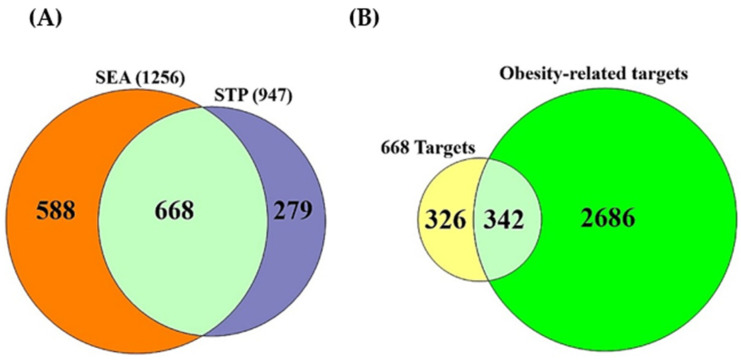
(**A**) The common 668 targets between SEA (1256) and STP (947). **(B**) The common 342 targets between the 668 targets and obesity-related targets (3028).

**Figure 3 cells-11-02903-f003:**
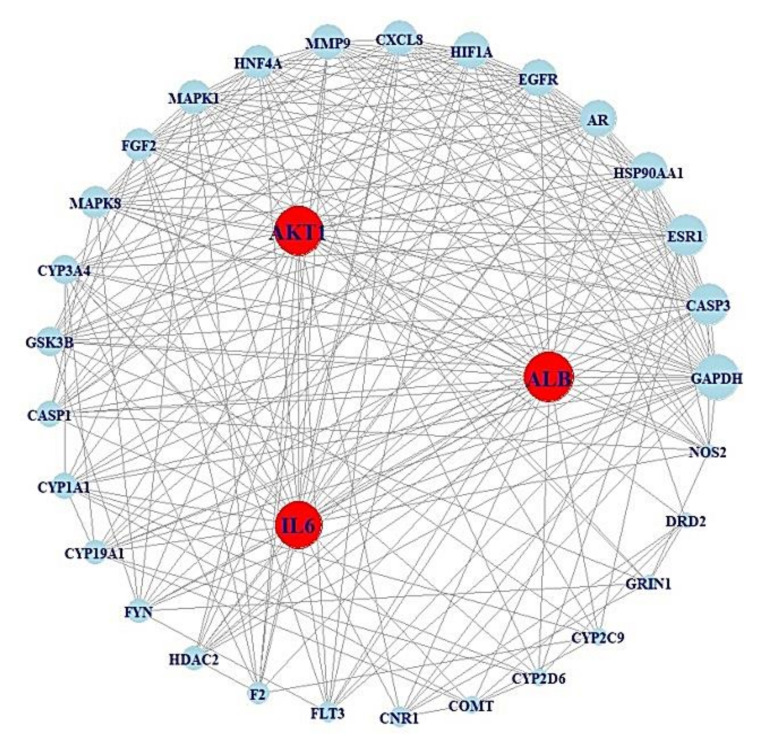
PPI networks (32 nodes, 254 edges) of the top 30% BC values from Figure 3.

**Figure 4 cells-11-02903-f004:**
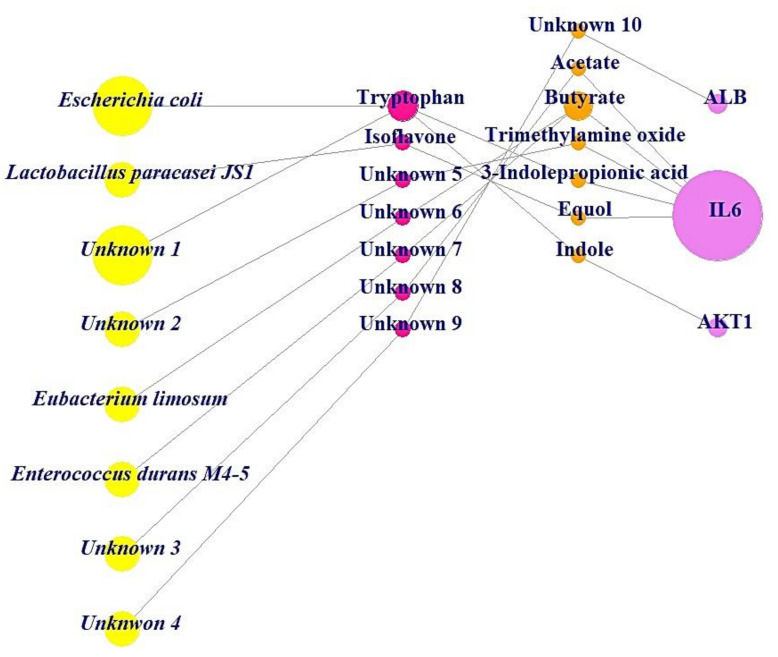
MSMT networks (25 nodes, 23 edges). Yellow circles: microbiota (probiotics); red circles: substrate (prebiotics); orange circles: metabolites (postbiotics); pink circle: target.

**Figure 5 cells-11-02903-f005:**
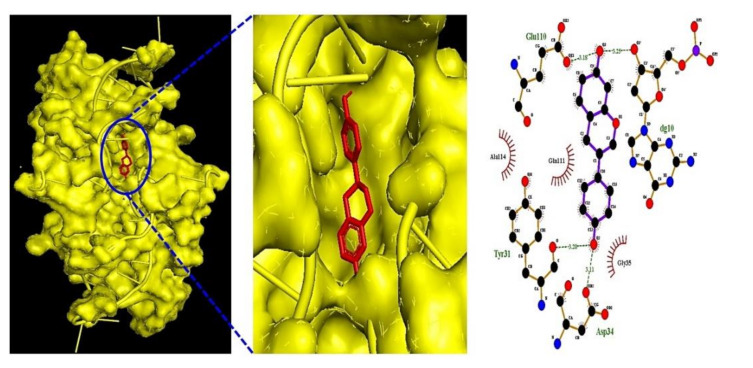
Equol–IL6 (PDB ID: 4NI9) complex on MDT.

**Table 1 cells-11-02903-t001:** List of databases used in the present study.

No.	Database	Brief Description	Utilization	URL
1	ADMETlab 2.0	A web-based platform to identify physicochemical properties of organic compounds	The pioneering of pharmcokinetics of organic compounds	https://admetmesh.scbdd.com/ (accessed on 4 June2022)
2	DisGeNET	A database of target–disease correlations	The pioneering of targets in response to diseases	https://www.disgenet.org/ (accessed on 1 June 2022)
3	gutMGene	Online database for identification of targets and metabolites from gut microbiota	The retrieval of targets and metabolites of gut microbes	http://bio-annotation.cn/gutmgene (accessed on 31 May 2022)
4	Online Mendelian Inheritance in Man (OMIM)	A collective compendium of human targets and diseases	The correlation of human targets and diseases	https://www.omim.org/ (accessed on 1 June 2022)
5	Similarity Ensemble Approach (SEA)	A database of targets related to compounds	The identification of potential targets on compounds	https://sea.bkslab.org/ (accessed on 31 May 2022)
6	String	A web-based tool to identify protein–protein interaction networks	The identification of network functional enrichment analysis	https://string-db.org/ (accessed on 1 June 2022)
7	SwissADME	A web-based tool for prediction of drug-like properties	The identification of physicochemical properties on compounds	http://www.swissadme.ch/ (accessed on 4 June 2022)
8	SwissTargetPrediction (STP)	A web server to explore targets from small molecules	The selection of targets on small molecules	http://www.swisstargetprediction.ch/ (accessed on 31 May 2022)
9	VENNY 2.1	A web-based tool for identification of overlapping elements	The identification and comparison of elements in a Venn diagram	https://bioinfogp.cnb.csic.es/tools/venny/ (accessed on 1 June 2022)

**Table 2 cells-11-02903-t002:** The degree of to 30% DC targets.

No.	Target	Degree of Centrality	No.	Target	Degree of Centrality
1	AKT1	156	54	ACLY	21
2	ALB	147	55	ALOX5	21
3	GAPDH	90	56	BACE1	21
4	CASP3	88	57	CSK	20
5	EGFR	85	58	CYP17A1	20
6	IL6	80	59	ELANE	20
7	ACE	71	60	F3	20
8	ESR1	71	61	HDAC6	20
9	CXCL8	65	62	MMP2	20
10	APP	61	63	ADCY5	19
11	EP300	59	64	ANPEP	19
12	AR	58	65	BCHE	19
13	HIF1A	58	66	CDK6	19
14	HSP90AA1	54	67	CHRNA4	19
15	CREBBP	51	68	CYP2C9	19
16	FGF2	46	69	HDAC4	19
17	MAPK1	42	70	HNF4A	19
18	ABCB1	39	71	IGFBP3	19
19	CASP8	39	72	INSR	19
20	GSK3B	39	73	ACE2	18
21	AHR	38	74	ADORA2A	18
22	CASP1	37	75	ADRB1	18
23	AKT2	36	76	FLT3	18
24	COMT	35	77	GSR	18
25	CYP3A4	35	78	HSPA1A	18
26	ACHE	34	79	AKR1C3	17
27	CNR1	34	80	BCL2A1	17
28	IL2	34	81	DRD2	17
29	ABCG2	33	82	NOS2	17
30	CTSB	33	83	NR3C1	17
31	NOS3	32	84	ADORA1	16
32	FYN	31	85	CHEK1	16
33	MAPK14	30	86	CTSL	16
34	ADRB2	29	87	CYP2D6	16
35	MMP9	29	88	FGF1	16
36	AKR1B1	27	89	GRIN1	16
37	ARG1	27	90	MAPT	16
38	CYP1A1	27	91	MCL1	16
39	F2	27	92	MET	16
40	CYP19A1	26	93	NFE2L2	16
41	ESR2	26	94	PPARA	16
42	IGF1R	26	95	AOC3	15
43	CCR2	25	96	CPB2	15
44	PPARG	25	97	REN	15
45	CD38	24	98	ALDH2	14
46	CDK1	24	99	ALOX15	14
47	CDK5	24	100	ERN1	14
48	CFTR	24	101	G6PD	14
49	CYP1A2	24	102	LGALS3	14
50	HDAC2	24	103	MMP3	14
51	MAPK8	24	104	NOS1	14
52	MPO	23	105	NR0B2	14
53	HDAC3	22	106	PTGS2	14

**Table 3 cells-11-02903-t003:** The degree of the top 30% BC targets from Table 1.

No.	Target	Betweenness Centrality	No.	Target	Betweenness Centrality
1	AKT1	1.000000	17	F2	0.121939
2	GAPDH	0.961904	18	AR	0.119210
3	EGFR	0.631284	19	GSK3B	0.111653
4	ALB	0.605009	20	DRD2	0.106535
5	CXCL8	0.564944	21	FYN	0.102145
6	ESR1	0.531729	22	NOS2	0.100364
7	IL6	0.519001	23	HDAC2	0.089496
8	CASP3	0.345339	24	FLT3	0.084114
9	HIF1A	0.344015	25	HNF4A	0.078172
10	CYP1A1	0.277903	26	GRIN1	0.068896
11	COMT	0.239681	27	CASP1	0.068437
12	HSP90AA1	0.227377	28	CYP19A1	0.067422
13	CYP3A4	0.210552	29	CYP2D6	0.064594
14	FGF2	0.198164	30	CNR1	0.063946
15	MAPK1	0.136887	31	CYP2C9	0.058081
16	MMP9	0.131470	32	MAPK8	0.057694

**Table 4 cells-11-02903-t004:** Molecular docking test of IL6 (PDB ID: 4NI9) and AKT (PDB ID: 3O96).

				Grid Box		Hydrogen Bond Interactions	Hydrophobic Interactions
Protein	Ligand	PubChem ID	Binding Energy (kcal/mol)	Center	Dimension	Amino Acid Residue	Amino Acid Residue
IL6 (PDB ID: 4NI9)	Equol	91469	−7.4	x = 11.213	x = 40	Glu110, Asp34, Tyr31	Gly35, Gln111, Ala114
				y = 33.474	y = 40		
				z = 11.162	z = 40		
	3-Indolepropionic acid	3744	−7.2	x = 11.213	x = 40	Arg16	Pro18, Gln17
				y = 33.474	y = 40		
				z = 11.162	z = 40		
	Trimethylamine oxide	1145	−3.6	x = 11.213	x = 40	N/A	N/A
				y = 33.474	y = 40		
				z = 11.162	z = 40		
	Butyrate	104775	−4.4	x = 11.213	x = 40	N/A	N/A
				y = 33.474	y = 40		
				z = 11.162	z = 40		
	Acetate	175	−3.8	x = 11.213	x = 40	N/A	Arg16
				y = 33.474	y = 40		
				z = 11.162	z = 40		
AKT1 (PDB ID: 3O96)	Indole	798	−5.2	x = 6.313	x = 40	Ser259	Asp262, Tyr417, Tyr263
				y = −7.926	y = 40		Gln414, His207
				z = 17.198	z = 40		

**Table 5 cells-11-02903-t005:** Physicochemical properties of equol.

No.	Compound	Lipinski Rules				Lipinski’s Violations	Bioavailability Score	Topological SurfaceArea (Å^2^)
		Molecular Weight	Hydrogen Bonding Acceptor	Hydrogen Bonding Donor	Moriguchi Octanol-Water Partition Coefficient			
		<500	<10	≤5	≤4.15	≤1	>0.1	<140
1	Equol	242.27	3	2	2.2	0	0.55	49.69

**Table 6 cells-11-02903-t006:** Toxicity profile of equol.

Parameter	Metabolite (Postbiotic)
	Equol
hERG blocker	Non-blocker
Rat oral acute toxicity	Negative
Eye corrosion	Negative
Respiratory toxicity	Negative
LD50 of acute toxicity	5.238 mg/kg

## Data Availability

All data generated or analyzed during this study are included in this published article (and its Appendix A).

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
