# Peer review of "Elucidation of Prebiotics, Probiotics, Postbiotics, and Target from Gut Microbiota to Alleviate Obesity via Network Pharmacology Study"

_cells, 2022, doi:10.3390/cells11182903_

Round 1

Reviewer 1 Report

The authors utilized network pharmacology to screen promising prebiotics, probiotics, postbiotics produced by guy microbiota and their targets. The results showed that equol converted from Isoflavone (prebiotics) via Lactobacillus paracasei JS1 can bind the most stably on IL6, which may be responsible for treatment of obesity. The manuscript is well written and straightforward. In introduction, if more information about network pharmacology and databases can be included, that would be helpful for general reader to understand the advantage and reliability of this approach. The quality of figures can be improved. E.g., it is hard to see the name of nodes in figure 3. What is the meaning of double lines in Table 3, 4 and 5? It is not necessary to use table for abbreviations. In discussion, more information about the function and mechanism of equol can be included to support the finding of this project. Also, except for Lactobacillus paracasei, there are many other intestinal bacteria can produce equol (J Nutr. 2010 Jul; 140(7): 1355S–1362S; Toxins (Basel). 2020 Jan 26;12(2):85; Front Microbiol.2022 May 20;13:901745). How can authors conclude Lactobacillus paracasei is the major bacteria responsible for equol production?

Author Response

The authors utilized network pharmacology to screen promising prebiotics, probiotics, postbiotics produced by guy microbiota and their targets. The results showed that equol converted from Isoflavone (prebiotics) via Lactobacillus paracasei JS1 can bind the most stably on IL6, which may be responsible for treatment of obesity. The manuscript is well written and straightforward.

  • Point 1 In introduction, if more information about network pharmacology and databases can be included, that would be helpful for general reader to understand the advantage and reliability of this approach.
  • Response 1: Thanks for great suggestion. We added up more information about network pharmacology in introduction section (Line 75-80)
  • Point 2 The quality of figures can be improved. E.g., it is hard to see the name of nodes in figure 3.
  • Response 2: Thanks for great suggestion. As a raw data, there are too many nodes. So, we attached as supplementary Figure S1. Additionally, we arranged the numbering of each figure.
  • Point 3 What is the meaning of double lines in Table 3, 4 and 5? It is not necessary to use table for abbreviations.
  • Response 3: Thanks for great suggestion. We deleted the double lines and corrected full name indicated in tables.
  • Point 4: In discussion, more information about the function and mechanism of equol can be included to support the finding of this project.
  • Response 4: Thanks for great suggestion. We added up more information in discussion section including Equol administration (Line 267-272).
  • Point 5: Also, except for Lactobacillus paracasei, there are many other intestinal bacteria can produce equol (J Nutr.2010 Jul; 140(7): 1355S–1362S; Toxins (Basel). 2020 Jan 26;12(2):85; Front Microbiol.2022 May 20;13:901745). How can authors conclude Lactobacillus paracasei is the major bacteria responsible for equol production?
  • Response 5: Thanks for great question and references. Above recommended the Three references, the results are based on in vitro The Lactobacillus paracasei JS1 we indicated is based on in vivo test. Additionally, our result is to show isoflavone (PubChem ID: 72304) as prebiotics, however, the three references are to show daidzein (PubChem ID: 5281708) as prebiotics in vitro test. It is a significant difference by comparison with our study.Thus, we think that Lactobacillus paracasei JS1 is better option due to proceeding to some degree in vivo test. The three references you recommended are very important information to verify the value in microbiome research in the future. Again, I appreciate your great recommendation.

Reviewer 2 Report

Comments and Suggestions for Authors

The hypothesis article ‚Elucidation of prebiotics, probiotics, postbiotics, and target from gut microbiota to alleviate obesity via network pharmacology study ‘by Ki-Kwang Oh et al. describes a network pharmacology approach to identify key targets in the microbiome responsible for obesity. The authors performed Meta Analyses. In a first approach, they identified microbiome-associated metabolites and associated them with obesity later on. They identified as key targets IL6, AKT1 and ALB. They promise a simple calculation of key targets that can be used to treat obesity.

In conclusion, this is a very interesting approach, targeting the available data in databases and performing meta-analysis and prediction models. Unfortunately, the manuscript offers very few explanations and discussions. Some figures should be replaced. The text should be improved, more explanation offered how they produced there data and more discussions included how they came to these conclusions.

Line 11: against…but also as crucial factors for obesity. This should be mentioned here.  

Line 27: remove one space.

Figure text 1: more explanations of the figure and the abbreviations used in the figure are required.

Figure 3: remove this figure. It is too crowded, cannot be read and should be described in the text or added as supplementary with much higher resolution or as a table but this way it does not make sense.  

Figure 4: same for figure 4.

Line 180: why do the authors know that these are unknown microorganisms? Isn’t a mixture of metabolites, contamination possible too?

Author Response

The hypothesis article ‚Elucidation of prebiotics, probiotics, postbiotics, and target from gut microbiota to alleviate obesity via network pharmacology study ‘by Ki-Kwang Oh et al. describes a network pharmacology approach to identify key targets in the microbiome responsible for obesity. The authors performed Meta Analyses. In a first approach, they identified microbiome-associated metabolites and associated them with obesity later on. They identified as key targets IL6, AKT1 and ALB. They promise a simple calculation of key targets that can be used to treat obesity.

  • Pont1: In conclusion, this is a very interesting approach, targeting the available data in databases and performing meta-analysis and prediction models. Unfortunately, the manuscript offers very few explanations and discussions. Some figures should be replaced. The text should be improved, more explanation offered how they produced there data and more discussions included how they came to these conclusions.
  • Response 1: Thanks for great suggestion. We added up more information in introduction and discussion section. On the whole, we revised the manuscript as you suggested. Thanks a lot.
  • Point 2: against…but also as crucial factors for obesity. This should be mentioned here.  
  • Response 2: Thanks for great suggestion. We revised it (Line 11-12).
  • Point 3: Line 27: remove one space.
  • Response 3: Thanks for great suggestion. We revised it.
  • Point 4: Figure text 1: more explanations of the figure and the abbreviations used in the figure are required.
  • Response 4: Thanks for great suggestion. We revised it and added up as full names.
  • Point 5: Figure 3: remove this figure. It is too crowded, cannot be read and should be described in the text or added as supplementary with much higher resolution or as a table but this way it does not make sense.  
  • Response 5: Thanks for great suggestion. As a raw data, there are too many nodes. So, we attached as Supplementary Figure S1. Additionally, we arranged the numbering of each figure.
  • Point 6: Figure 4: same for figure 4.
  • Response 6: Thanks for great suggestion. We changed it as Supplementary Figure S2.
  • Point 7: Line 180: why do the authors know that these are unknown microorganisms? Isn’t a mixture of metabolites, contamination possible too?
  • Response 7: Thanks for great question. The meaning of “unknown” is that it does not know what components are. For instance, the “Unknown 1” (unknown gut microbiota) might intake “Unknown 5” (unknown probiotics) which can be converted into “Trimethylamine oxide”. Thus, if some researchers confirm the elements, there will be spectacular results. We give clues to demonstrate with quick methodology. Again, we appreciate your valuable question.

Reviewer 3 Report

 It is a very conscientious work and with great bioinformatic work behind it, however, I have some criticisms and doubts for the authors.

 Abstract:

“ The metabolites produced by gut microbiota have been reported as crucial agents against  obesity, but its key target(s) have not been revealed completely in complex microbiome system.  Hence, this study aimed to decipher promising prebiotics, probiotics, postbiotics, more importantly, key target(s) …”

I do not understand this link

Introduction:

In my view, the introduction is not sufficiently well argued and the different points are not well connected. Neither are details given about the microbiota-obesity relationship nor is the background related to this topic sufficiently exposed.

 What is the objective of this work? I understand that it is to demonstrate the hypothesis described, but I think that the objective should be better described at the end of the introduction as well as the benefits of the results obtained.

What criteria were used to select the metabolites produced by gut microbiota?

Why those and not others?

A conventional flow chart would be more explanatory.

 In the end, after reading and reading the paper, it is still not clear to me how the conclusion is reached:

 "Taken together, our study shows that Isoflavone (Prebiotics), Lactobacillus paracasei JS1 (Probiotics), Equol (Postbiotics), and IL6 (Target) are the most crucial components against obesity in current microbiome research."

 The authors should make an effort to justify and detail each step.

Are there studies that, at least in part, ratify the bioinformatics results? This aspect should be indicated.

Author Response

It is a very conscientious work and with great bioinformatic work behind it, however, I have some criticisms and doubts for the authors.

 Abstract:

  • Point 1: “ The metabolites produced by gut microbiota have been reported as crucial agents against  obesity, but its key target(s) have not been revealed completely in complex microbiome system.  Hence, this study aimed to decipher promising prebiotics, probiotics, postbiotics, more importantly, key target(s) …” I do not understand this link
  • Response 1: Thanks for great question. Our study is to reveal the four main components in microbiome research. The prebiotics is a precursor taken by probiotics (gut microbiota) to convert into significant therapeutic metabolites (defined as postbiotics). Consequently, the metabolites (defined as postbiotics) might exert therapeutic benefits by binding key target(s).

Introduction:

  • Point 2: In my view, the introduction is not sufficiently well argued and the different points are not well connected. Neither are details given about the microbiota-obesity relationship nor is the background related to this topic sufficiently exposed.
  • Response 2: Thanks for great suggestion. As your suggested, we added up more their relationships more detail. (Line 52-59).
  • Point 3: What is the objective of this work? I understand that it is to demonstrate the hypothesis described, but I think that the objective should be better described at the end of the introduction as well as the benefits of the results obtained.
  • Response 3: Thanks for great suggestion. We added up more detailed information in end of introduction to explain the objective of our study (Line 89-95).
  • Point 4: What criteria were used to select the metabolites produced by gut microbiota?
  • Response 4: Thanks for great question. All gut metabolites enrolled in gutMGene database were selected, the method was commented on methodology step1. Finally, we evaluated the drug-likeness properties via in silico screening like SwissADME platform.
  • Point 5: Why those and not others?
  • Response 5: Thanks for great question. On MDT, we selected the highest affinity on IL6 (A key target). A metabolite with lowest binding energy is just Equol.
  • Point 6: A conventional flow chart would be more explanatory.
  • Response 6: Thanks for great suggestion. I added up more explanation.
  • Point 7: In the end, after reading and reading the paper, it is still not clear to me how the conclusion is reached:

 "Taken together, our study shows that Isoflavone (Prebiotics), Lactobacillus paracasei JS1 (Probiotics), Equol (Postbiotics), and IL6 (Target) are the most crucial components against obesity in current microbiome research."

  • Response 7: Thanks for great question. As above answered in Response 1, we integrated the four key components to understand microbiome study and analyzed systemic relationship. We wish that it is helpful to understand our approach.
  • Point 8:  The authors should make an effort to justify and detail each step.
  • Response 8: Thanks for great suggestion. We added up more contents to justify in introduction and method sections.
  • Point 9: Are there studies that, at least in part, ratify the bioinformatics results? This aspect should be indicated.
  • Response 9: Thanks for great suggestion. We added up the information in Step3. Again, I appreciate your valuable advice.

Reviewer 4 Report

In the manuscript titled "Elucidation of prebiotics, probiotics, postbiotics, and targets from gut microbiota to alleviate obesity via network pharmacology study" the authors Ki-Kwang Oh and colleagues have hypothesized that target(s) with the highest betweenness centrality (BC) degree in PPI networks may be potential treatment targets. Key postbiotics are metabolites that are most stably bound to a key target in a molecular docking test (MDT). A microbe that produces postbiotics is also considered a key probiotic. Regarding the present manuscript, I would like to make a few comments.

-Thank you for giving me the opportunity to revise the hypothesis article. This is my first experience with this type of article, and I have enjoyed reading it. The authors used a new database that was published in January 2022. Low resolution is the main problem with the different figures.

-In addition to the websites that the authors have used, the references should also be included.

-A discussion of the quality requirements should be included in the text. For instance, why the degree of centrality should be greater than 30% 

-Three genes were selected at the end of the study, and it is possible that the number of raw data that is selected is an important consideration in this type of research.

-The ADME study is really impressive. Thank you for sharing it

-There should be more discussion than on half a page. The authors could discuss the validity of the workflow, how it was determined, quality issues, and the final outcome, as well as the rationality of the outcome.

Author Response

In the manuscript titled "Elucidation of prebiotics, probiotics, postbiotics, and targets from gut microbiota to alleviate obesity via network pharmacology study" the authors Ki-Kwang Oh and colleagues have hypothesized that target(s) with the highest betweenness centrality (BC) degree in PPI networks may be potential treatment targets. Key postbiotics are metabolites that are most stably bound to a key target in a molecular docking test (MDT). A microbe that produces postbiotics is also considered a key probiotic. Regarding the present manuscript, I would like to make a few comments.

  • Point 1: Thank you for giving me the opportunity to revise the hypothesis article. This is my first experience with this type of article, and I have enjoyed reading it. The authors used a new database that was published in January 2022. Low resolution is the main problem with the different figures.
  • Response 1: Thanks for great suggestion. As suggested, I made at maximum resolution.
  • Point 2: In addition to the websites that the authors have used, the references should also be included.
  • Response 2: Thanks for great suggestion, we added up the references about using websites.
  • Point 3: A discussion of the quality requirements should be included in the text. For instance, why the degree of centrality should be greater than 30%
  • Response 3: Thanks for great suggestion. Overall, we added up more contents in the discussion section. Also, we explained the reason why 30% BC used in the study.
  • Point 4: Three genes were selected at the end of the study, and it is possible that the number of raw data that is selected is an important consideration in this type of research.
  • Response 4: Thanks for great suggestion. Your opinion is exact. We simplified it via MDT. We think that three targets can be significant elements against obesity. Again, we appreciate your opinion.   
  • Point 5: The ADME study is really impressive. Thank you for sharing it.
  • Response 5: Thanks for great opinion. Again, we appreciate your positive assessment.
  • Point 6: There should be more discussion than on half a page. The authors could discuss the validity of the workflow, how it was determined, quality issues, and the final outcome, as well as the rationality of the outcome.
  • Response 6: Thanks for great suggestion. We combined the contents in discussion section.

Round 2

Reviewer 3 Report

Thanks for your answers. My questions have been answered for the most part.

I have one concern: I believe that this point still needs a little more clarification.

It would also be convenient to introduce it in the text

 Point 1: “ The metabolites produced by gut microbiota have been reported as crucial agents against obesity, but its key target(s) have not been revealed completely in complex microbiome system. Hence, this study aimed to decipher promising prebiotics, probiotics, postbiotics, more importantly, key target(s) …” I do not understand this link

Response 1: Thanks for great question. Our study is to reveal the four main components in microbiome research. The prebiotics is a precursor taken by probiotics (gut microbiota) to convert into significant therapeutic metabolites (defined as postbiotics). Consequently, the metabolites (defined as postbiotics) might exert therapeutic benefits by binding key target(s).

Author Response

Reviewer 3:

Thanks for your answers. My questions have been answered for the most part.

  • Point 1: I have one concern: I believe that this point still needs a little more clarification. It would also be convenient to introduce it in the text Point 1: “The metabolites produced by gut microbiota have been reported as crucial agents against obesity, but its key target(s) have not been revealed completely in complex microbiome system. Hence, this study aimed to decipher promising prebiotics, probiotics, postbiotics, more importantly, key target(s) …” I do not understand this link. Response 1: Thanks for great question. Our study is to reveal the four main components in microbiome research. The prebiotics is a precursor taken by probiotics (gut microbiota) to convert into significant therapeutic metabolites (defined as postbiotics). Consequently, the metabolites (defined as postbiotics) might exert therapeutic benefits by binding key target(s).
  • Response 1: Thanks for great suggested. As you suggested, we added up detailly the contents between line 59 and 64 in introduction section. Again, we appreciate your kind consideration.

Reviewer 4 Report

Thanks to the authors for considering my previous comments. All of those comments have been addressed and further comments are not required.

Author Response

Reviewer 4:

  • Point 1: Thanks to the authors for considering my previous comments. All of those comments have been addressed and further comments are not required.
  • Response 1: Thanks for your positive assessment. Thanks a lot.